# Epidemiological Features of the Highly Pathogenic Avian Influenza Virus H5N1 in a Densely Populated Area of Lombardy (Italy) during the Epidemic Season 2021–2022

**DOI:** 10.3390/v14091890

**Published:** 2022-08-26

**Authors:** Silvia Bellini, Alessandra Scaburri, Erika Molica Colella, Monica Pierangela Cerioli, Veronica Cappa, Stefania Calò, Marco Tironi, Mario Chiari, Claudia Nassuato, Ana Moreno, Marco Farioli, Giuseppe Merialdi

**Affiliations:** 1Istituto Zooprofilattico Sperimentale della Lombardia ed Emilia-Romagna, Via Bianchi 9, 25124 Brescia, Italy; 2Direzione Generale Welfare di Regione Lombardia, Unità Organizzativa Veterinaria, Piazza Città di Lombardia, 20124 Milan, Italy

**Keywords:** H5N1, avian influenza, poultry, risk factors, densely populated areas

## Abstract

In the last two years, there have been three major epidemic seasons in the territory of the European Union and the HPAI epizootic in 2021–2022 is the most severe in recent history. In Italy, the disease was introduced to dense poultry areas with serious economic consequences for the entire sector. In Lombardy, the analysis of the risk factors was carried out, also taking into account the density of domestic birds. In the most affected areas, 66.7% of the outbreaks occurred in the areas with the highest poultry density and the likelihood of an outbreak occurring increased with an increase in the density of birds per km^2^. In cells 10 × 10 km with a density greater than 10,000 birds/km^2^, the probability of outbreak occurrence was over 66.7%. The provinces involved in the last epidemic were the same involved in previous epidemics and, given the risk factors present in the area, it is plausible that the risk remains high also for future epidemic seasons. Therefore, to avoid the repetition of similar events, certain control measures shall be strengthened and vaccination considered as a complementary tool for the control of HPAI virus in risk areas.

## 1. Introduction

The 2021-2022 Highly Pathogenic Avian Influenza (HPAI) epizootic is considered the largest epidemic to have occurred so far in Europe [1]. At present, the epidemic is still ongoing in some European countries with cases reported in poultry and wild birds (Figure 1). It affected 36 European countries with a total of 2398 outbreaks in poultry, 46 million birds killed in affected establishments, 168 captive bird detections and 2733 HPAI cases in wild birds. Two detection peaks of HPAI A (H5) virus were observed in wild birds during the epidemic season: the first was in early November 2021, coinciding with the arrival of waterfowl at wintering sites in Europe, the second was recorded in mid-January 2022. When considering HPAI virus in poultry, France, with 68% of the outbreaks, was by far the most affected European country. All HPAI viruses (H5Nx) characterized by October 2021 belong to the clade 2.3.4.4b and, unlike the previous epidemic seasons, H5N1 resulted the predominant subtype identified [1].

Migratory wild birds, especially waterfowls, are the natural host and reservoir of avian influenza viruses. Depending on the virus strain and the species of bird, the virus can be harmless or fatal to the wild bird. The observed persistence of HPAI (H5) virus in wild birds since the 2020–2021 epidemic indicates that it may have become endemic in wild bird populations in Europe [1]. Moreover, recent research on various species of dabbling ducks in Italy [2] showed a high prevalence of HPAI A (H5) infection in winter in geographical areas where no dead birds were detected, indicating that high infection pressure in the environment is possible also in the absence of wild bird mortality. This is in line with a recent study conducted in the UK in which post-mortem examinations were performed on multiple wild bird species diagnosed with HPAI A (H5) infection in 2020–2021. Some of these birds had serious influenza lesions while others showed no obvious macroscopic abnormalities [3]. It is worth remembering that the presence of dead wild birds is an early indicator of the presence of the infection in a particular area, which triggers the adoption of control measures in poultry. Therefore, the absence of mortality of wild birds can delay the adoption of control measures in poultry with serious consequences, especially in densely populated areas.

Northern Italy (Veneto and Lombardy region) was involved in this last epidemic from October 2021 to February 2022, the virus was introduced in areas with a high density of poultry. In the same areas there are wetlands and migratory birds, which make an ideal ecological habitat for the maintenance and spread of influenza viruses. The poultry sector in Lombardy together with Veneto and Emilia Romagna represents 63% of national production [4]. In Lombardy, there were 1269 commercial farms with 26 million birds at 23 November 2021. The highest concentration of intensive farming, representing about 80.9% of the poultry stock of the region, is distributed in the provinces of Brescia (457 farms), Mantua (207 farms) and Cremona (173 farms), where about 21 million birds are reared; indeed, these are the three provinces affected by the H5N1 epidemic in 2021–2022.

In Italy, the first H5N1 case in domestic birds was confirmed in October 2021 in a turkey farm in the Veneto region and, about a month later, the virus was also identified in poultry in Lombardy. The first occurrence of the virus in wild birds was reported in Lombardy at the end of October 2021. The infection was identified in Brescia and, in the same province, the same virus was found in mid-November in the first infected flock. A total of 315 H5N1 outbreaks were detected in northern Italy in less than four months, of which 60 (19.04%) were reported in Lombardy. Most of the outbreaks occurred in areas with a high density of poultry and the epidemic was characterized by a rapid spread of the infection. In Lombardy, the affected area is contiguous to the densely populated area of Veneto, the same affected by previous HPAI epidemics. In Italy, these areas of infection are among those considered to be at high risk of the spread of avian influenza (Zone B of Annex II to the DGSAF note ref. 29,049 of 20 November 2019) also in consideration of the wet-lands present in the areas which represent a further risk factor for the presence of wild birds that directly, or more often, indirectly, can infect domestic birds. Initially two spatiotemporal clusters of infection were identified in Lombardy, with no epidemiological links between the two (Figure 2). The first occurred in Brescia, where the virus had already been identified in the wild, the second in the province of Mantua, on the border with Veneto, where the disease was already spreading to domestic birds. Subsequently, the disease spread locally, affecting the high-risk area of transmission with a peak in mid-December 2021 (Figure 3). The virus detected in Veneto and Lombardy was characterized as subtype H5N1 and the investigation conducted by the European Reference Laboratory for avian influenza indicated that there have been at least seven distinct introductions of viruses from wild birds into the domestic bird population, with several outbreaks caused by viruses showing a high genetic relationship [1], which could be explained by the secondary spread of HPAI virus between poultry establishments.

Among the commercial establishments affected in Lombardy, the vast majority were large commercial establishments for laying hens (20/60) and fattening turkeys (20/60). Other farms kept broilers (9/60), domestic ducks for fattening (4/60), guinea fowl (1/60) and breeding chickens (1/60). The remainder were family flocks of poultry (5). Mortality or symptoms were reported for all the outbreaks. However, in the outbreaks (11) identified following monitoring in risk areas, which included both sample checks on live birds and on deaths, in two cases the swabs taken from the carcasses were positive and those taken from live birds were still negative. This seems to indicate that in the early stages of infection, sampling of the dead (passive surveillance) is more effective for identifying the disease early.

The most recent outbreaks of the epidemic in poultry were detected in both regions in early January 2022. The eradication measures provided for in the Commission Delegated Regulation (EU) 2020/687 were promptly adopted, including culling all poultry on infected farms, restricting the movement of poultry, vehicles and staff [5]. Measures also included the pre-emptive slaughter of flocks that had been in contact with infected flocks or were in close proximity to infected flocks. There were difficulties coping with the disposal of such large numbers of poultry carcasses, given that two regions with a high density of poultry and large farms were involved at the same time. At the end of the epidemic, in Lombardy there were about 5 million birds culled, 4.5 million birds in outbreaks and 500,000 during preventive slaughter with estimated direct costs (birds killed, products destroyed, etc.) of around EUR 16 million. However, despite the eradication of the disease in domestic birds, the virus continued to spread in the wild up to the end of January.

Factors determining the spread of an infection include the frequency and type of contact among farms in the period following infection before the infection is detected and the farm is isolated as well as the density of farms in the area surrounding the infected farm [6,7]. These factors play a pivotal role in the development of an epidemic, especially when the pathogen is introduced into a densely populated area. At the time that H5N1 was introduced in Lombardy, the farms in the affected areas were already under a restriction regime due to the presence of the disease on the border of the region, in the province of Verona (Veneto). In both regions, most of the outbreaks were in close proximity and occurred in areas with high density of poultry where the epidemic was characterized by the rapid spread of the infection. The occurrence of multiple short-distance outbreaks is in line with what is expected in densely populated areas due to neighborhood infections [8].

Proximity between farms is commonly used as a risk factor in epidemiological analysis and in the control of epidemic diseases in livestock, even in the absence of accurate information on the spread of diseases between farms [7]. Indeed, in densely populated areas it is often difficult to determine how the disease spreads between farms over short distances, though different transmission routes have been hypothesized involving people, airborne transmission, flies and rodents [9].

It has also been shown that the risk of infection increases as neighboring flocks grow in size. In fact, flock size is considered a risk factor because, as the number of susceptible birds increases, the likelihood of disease transmission through contact also increases. Furthermore, it has also been found that the radial dispersion of the virus is more significant than dispersion along the road [8]. This is the rationale for including preventive slaughter in EU legislation as a control option for major epidemic diseases.

The aim of this paper is to present certain epidemiological features of the H5N1 epidemic 2021–2022 in Lombardy with an analysis of the main risk factors identified in the outbreaks during the epidemiological investigations. Considering the high density of poultry in the areas where the outbreaks occurred, the avian flu epidemic was analyzed taking into account the density of domestic birds in the most affected areas. Indeed, 66.7% of the outbreaks occurred in areas with the highest poultry density in the region. As the density of birds per km^2^ increases, the probability of an outbreak also increases. In cells with a density greater than 10,000 birds/km^2^, the probability of outbreak occurrence was higher than 66.7%. Finally, some concluding remarks and lessons to be learned from this last epidemic were reported.

## 2. Materials and Methods

For the data processing analyzed in the study, three sources of information were used:Epidemiological investigation of the outbreaks;National Animal Health Information System (SIMAN): Complete list of highly pathogenic avian influenza outbreaks (commercial, backyard and wild bird);Regional Database (BDR): list of poultry farms operating in the region with personal information and geographical coordinates as of 23 November 2021, 9 December 2021, 30 December 2021 and 11 January 2022.

Epidemiological investigations were conducted using the specific form provided by the National Reference Centre for avian influenza, which collects animal registration details, anamnestic and clinical information, risk factors and other information useful for tracing other outbreaks. All suspected cases of HPAI virus in poultry were investigated pursuant to Commission Delegated Regulation (EU) 2020/689 which lays down the rules for surveillance, eradication program, and disease-free status for certain listed and emerging diseases [10].

Data extracted from SIMAN and BDR were mapped using QGis software [11]. A 10 × 10 km grid was also created, with a total of 189 cells, which was superimposed on the regional territory. The farms found in each cell were analyzed by calculating the density of birds per km^2^ and the outbreaks.

Using the data on density, the Welch test was used to verify whether the average density value as of 23 November 2021, 9 December 2021, 30 December 2021 and 11 January 2022 deviated significantly between cells in which there has been at least one outbreak versus those where there has been no outbreak. The association between the presence of an outbreak in the cell (outcome) compared to the density as of 23 November 2021 was estimated using a logistic regression model. This model allows us to estimate the risk (with relative 95% confidence interval) of an outbreak occurring in a cell. Finally, the probability that an outbreak would occur in a cell was calculated according to the following density thresholds per km^2^: 100, 500, 1000, 5000 and 10,000. Fisher’s exact test was used to verify the existence of a relationship between biosecurity measures and the presence of outbreaks in densely populated cells.

The statistical analysis was performed using the software R version 3.6.1 (R core Team: Vienna, Austria, 2018, accessed on 15 July 2022) [12].

## 3. Results

### 3.1. Outbreaks Detected during the HPAI 2021–2022 Epidemic in the Lombardy Region

A total of 64 outbreaks were detected during the HPAI 2021–2022 epidemic in Lombardy, of which four wild birds and five backyards (Table 1). For the purpose of the study, the 54 outbreaks in commercial farms were considered: 26 in the province of Brescia, 26 in the province of Mantua and 2 in the province of Cremona. The outbreak detected in Pavia was excluded from this analysis because was not part of the same epidemic event, and it was attributable to different viral introduction.

A larger proportion of the farms involved were large establishments (i.e., with more than 10,000 birds). In total, 5 outbreaks occurred in farms with more than 100,000 birds (224,957 birds in the largest farm), 11 farms were in the category of 50,001–100,000 birds, 34 in the category of 10,001–50,000 birds and five flocks fewer than 10,000 birds.

### 3.2. Risk Factors Identified in the Outbreaks of the 2021–2022 Epidemic in Lombardy

Based on the epidemiological investigations carried out, it emerged that all the establishments involved in the avian flu epidemic were exposed to one or more risk factors listed in Table 2. Almost all of the farms were within the protection or surveillance zones for other outbreaks in the Lombardy region and/or the neighboring Veneto region. Furthermore, environmental risk factors such as the proximity of water sources and the sighting of wild birds (turtledoves, herons, ibises, egrets, mallards, pigeons, cormorants) in the vicinity or in some cases within the farm areas were reported in all outbreak sites. In each outbreak site, there were nearby irrigation ditches; in addition, three farms in the province of Brescia (Milzano, Pavone del Mella) are located near the Mella river and one farm in the municipality of Pozzolengo was near Lake Garda and a small lake for live decoys. As concerns the Mantua outbreaks, two farms were located in the vicinity of live decoy lakes and one near the River Chiese.

Management risk factors were also considered, such as belonging to the same poultry company, since there may also be indirect contacts between these farms that are not always easy to document. The majority of the affected farms (98%) belonged to poultry companies, 35% were part of the two biggest companies in Italy, which in these areas (Brescia, Mantua and Verona) have found optimal conditions for growth. In consideration of the large number of birds within the outbreak sites and the population density, it was also examined the type of ventilation in relation to the type of birds farmed. It turned out that all of the layer farms that were the site of an outbreak had forced ventilation, which may have contributed to generating locally a high viral load in the environment. It is significant to report that in Lombardy all the outbreaks occurred in the plains (Po Valley) [13] and the meteorological conditions in the period of maximum spread of the infection were characterized by a thick fog in the affected areas; all conditions that could have favored a stagnation of the virus in the environment.

As shown in Table 2, epidemiological investigations also revealed outbreaks linked by indirect contact involving staff and vehicles. In the province of Brescia, family ties were identified between the owners of two farms where outbreaks occurred; indeed, up to the date of the suspected infection, the two farms were run jointly, in addition to being very close to one another (500 m). In the province of Mantua, two outbreaks identified in the same municipality involved farms run by the same farmer. In addition, another farm in the province of Mantua was linked to two outbreaks in Veneto: one was run by the same farmer and the disease reached the other by indirect contact via vehicles. Indirect contact via vehicles was also reported for two other outbreaks in Mantua: one linked to an outbreak in Veneto and the other to an outbreak in Brescia.

It is worth remembering that, at the time of the onset of the epidemic in Lombardy, the farms in the region in the area at risk had already been placed under restrictions due to the presence of the infection in areas with a high risk of transmission in the neighboring province of Verona (Veneto), also in consideration of the operational links between the two areas. In these circumstances, the only movements permitted for the farms in these areas were those necessary for maintaining the conditions of animal welfare. For this reason, in 26 outbreak sites in the province of Mantua that had the feed unloading point inside the farm, an in-depth analysis of indirect contact via the feed delivery vehicles (Table 3) in the period at risk was carried out. The risk period was defined considering the flock incubation period of 14 days, as established by WHOA Terrestrial Code for high pathogenicity avian influenza viruses [14]. From the analysis of the data, it emerged that, in the period at risk, 19 of the 26 outbreaks had indirect contact via the feed delivery vehicle with farms that were already the site of outbreaks. However, this was considered a weak element of risk, since these vehicles had to undergo rigorous biosecurity measures before entering the farm and in no case a vehicle had immediately prior contact with an outbreak site before entering a farm. This means that, between the two possible high-risk contacts, there were other contacts with farms that did not become an outbreak or intermediate trips to the feed mill where the vehicles were cleaned and disinfected.

In Table 3, the outbreaks found in high-density areas are indicated in bold and, in Figure 4, letters are used to show the outbreak sites that had high-risk contact via the feed delivery vehicles.

### 3.3. Analysis of Outbreaks in Relation to Poultry Density Data

Given the high density of farms in the areas where outbreaks were detected and the characteristics of spread of the disease in the affected areas, it was decided to analyze the epidemic also as a function of the density of poultry in the most affected areas, given that it is known that the density of susceptible animal species is one of the elements influencing the spread of a disease.

The initial dataset consists of 1269 intensive establishments operating as at 23 November 2021, as extracted from the BDR. A total of 271 intensive establishments were found in the cells in which at least one outbreak has occurred. For each cell involved, the density of the birds per cell was calculated (per km^2^) as of 23 November 2021, 9 December 2021, 30 December 2021 and 11 January 2022 and it was observed that 66.7% of the outbreaks in commercial farms (36/54) took place in the most densely populated (density ≥ 7138 head per km^2^) as of 23 November 2021, corresponding to cells 77, 96, 97, 100, 117 and 118.

Table 4 shows the information for the cells where outbreaks occurred in commercial farms and it emerged that cell 96, which is located in the province of Brescia, had the greatest number of outbreaks (14 out of 54 intensive farming establishments) and it was found to have the highest density of birds both on 23/11 and on 09/12, compared to the whole of Lombardy. Epidemiological investigation of the outbreaks in this cell revealed that all outbreaks had at least two combined risk factors. All the outbreaks fell within at least one restriction zone for other outbreaks; two of these were also linked by the farmers being related through family; 86% of the outbreaks in this cell belonged to the same poultry companies. Almost all of these outbreak sites (93%) were exposed to water sources next to the farm with the presence of wild animals and 78% had forced ventilation. It is worth remembering that 9 farms out of the 14 outbreak sites in this cell had already been the site of an outbreak in the HPAI epidemic in 2017/2018.

Cell 97, adjacent to cell 96, was affected by 11 outbreaks and reported a density of 9141.88 birds per km^2^ at 23 November 2021. The epidemiological investigations revealed that the outbreak farms in this area had the same risk factors found in the outbreak sites of cell 96, with the exception of the family relationship between farmers.

Instead, cell 74, which fell in the province of Cremona but is adjacent to the most affected cells in Brescia (96 and 97, which account for 25 out of 54 outbreaks, equal to 46.3%), was the area of the only two outbreaks that occurred in the province of Cremona at the start of the epidemic (25 November 2021 and 1 December 2021), 3.7% of the total (2/54). In cell 74, there was no further spread of the infection and the population density was 8962.05 birds per km^2^ lower than cell 96 and 3244.86 heads per km^2^ lower than cell 97.

The situation in these cells was described in detail insofar as, although the first outbreaks were all close to the first case of H5N1 found in wild birds in Seniga (Brescia), there was no further spread of the infection in Cremona (cells 74 and 75), unlike what happened in Brescia (cells 96 and 97).

Figure 5 shows the distribution of the outbreaks in the cells concerned: the darker color corresponds to a greater density. As can be seen, most of the outbreaks developed in the cells with the highest density.

### 3.4. Analysis of the Association between the Presence of Outbreaks and Bird Density

Table 5 shows that the average density in the cells affected by outbreaks is statistically higher than in the cells free from outbreaks, for all the dates considered. In the table is shown that the average density decreased over time in cells with outbreaks following the application of disease stamping-out and control measures, while it remained stable in cells without outbreaks.

The regression model reported in Table 6 confirmed the Welch test results and also shows how the probability of an outbreak in a cell varied with variation in density. In particular, when the cell was not very densely populated (fewer than 100 birds per km^2^), the probability of an outbreak in the cell was under 2%; in contrast, when the density was greater than 10,000 birds per km^2^, the probability of an outbreak occurring was over 66.7%.

For the sake of completeness, it was also verified whether there were substantial differences in the application of biosecurity measures between the cells with or without outbreaks, by verifying the checks carried out in 2021 by the Veterinary Services. The data were analyzed using Fisher’s exact test and it emerged that the degree of application of biosecurity did not seem to have affected the presence of outbreaks in the cells considered; in fact, the *p*-value was not statistically significant (*p* = 0.25).

## 4. Discussion

According to the genotyping data published by the National Reference Centre for avian influenza, the recent H5N1 epidemic was characterized by multiple introductions of the infection via transmission from wild to domestic birds with the subsequent rapid spread of the infection in domestic birds [15]. In fact, the disease was introduced in areas dense with poultry, the same already affected by previous epidemics in northern Italy, where all the risk factors for spreading the infection, typical of these areas, are present.

In Lombardy, the first report of the disease was made at the end of October 2021, in wild birds hunted in the province of Brescia, the same virus was then identified in mid-November in the first domestic outbreak in Brescia. The latest reports of H5N1 were in wild birds in the provinces of Mantua and Cremona, when the disease had already been eradicated from commercial farms. This implies that the persistence of the virus in wild birds continued to pose risks for the poultry industry in the area.

Most of the outbreaks detected in the provinces of Brescia, Mantua and Cremona (96%) were identified in the areas under restrictions following other outbreaks; in the province of Mantua also in areas under restrictions following outbreaks in Veneto. Indeed, the province of Mantua is on the border with Verona (Veneto), which was the province most affected by the disease in the recent epidemic. Within the Lombardy areas at highest risk of transmission, the disease affected the areas with the highest density of poultry (65.5% of outbreaks). The majority of the outbreaks (98.1%) occurred in farms that are part of poultry companies, of which 35% belonged to two companies that were also the most affected by the disease in Veneto. Some outbreaks had also been the sites of infection in previous influenza epidemics.

The statistical analysis confirmed what was already evident from the geographical distribution of the outbreaks: as the density of birds per km^2^ increased, the likelihood of an outbreak occurring also increased. In cells with a density over 10,000 birds per km^2^ the probability of an outbreak was over 66.7%. Furthermore, the risk of infection from a single flock was also related to the size of the farm, and in the affected area, the size of the establishments involved was quite large: the average outbreak site housed 46,618 birds while the maximum farm size was 224,957 birds. In this situation, the risk of infection increases also in neighboring flocks.

In the recent H5N1 epidemic in Lombardy, the relevance of population density in the development of the epidemic was confirmed and, considering the characteristics of the production system in the area, it is plausible that the risk of spreading will remain high in the next epidemic seasons. In fact, in the area between Verona, Mantua and Brescia, there is a strong vocation for agriculture and, over time, optimal conditions have been created for the development of poultry farming. At the same time, these areas also feature suitable habitats for wild bird species that are effective in the transmission of influenza viruses. Therefore, to avoid the repetition of similar events, effective tools must be identified to enable poultry farming to coexist with these risk situations. The control strategies for such areas should be better targeted, also by establishing a limit for the susceptible population density and applying farm biosecurity measures suitable for such areas. Indeed, in the absence of vaccination, biosecurity is key to preventing the disease from being introduced to and spreading on farms and is thus also crucial to minimizing the impact of the epidemic. To reduce the risk of secondary transmission, early detection needs to be improved also in poultry, not only in wild birds. This could be achieved by identifying in peacetime the flocks at the highest risk of introducing the infection, with a view to upping passive surveillance immediately when the disease is identified in the wild. However, it is worth mentioning that a high prevalence of HPAI A (H5) infection has been demonstrated in areas where no dead birds have been detected. This means that in such circumstances, in order to enhance early warning, passive surveillance must be integrated with active surveillance in selected strategic areas. Furthermore the rendering capacity needs be improved in these areas, also by identifying alternative disposal methods, such as composting, given that the rendering process in the recent H5N1 outbreak was the bottleneck that significantly slowed down the application of outbreak control measures.

## 5. Conclusions

In the last two years, three major epidemic seasons have occurred in the European Union, and the last HPAI epizootic is the most severe in recent history. It appears that the control of HPAI virus has become a more complex issue than in the past, particularly when the disease is introduced into densely populated areas. Therefore, the Council of the European Union, faced with the devastating consequences of the recent HPAI epidemic in the European poultry sector, recently adopted conclusions on a strategic approach for the development of vaccination as a complementary tool for the prevention and control of HPAI epidemic in the EU [16]. It should be noted that the World Organisation for Animal Health (WOAH) and the EU legislation already provide for vaccination against HPAI virus. Accordingly, the Council of the European Union is required to intensify efforts to develop vaccination strategies for the prevention and control of HPAI virus, to be implemented on areas, species and farming practices that are at risk.

## Figures and Tables

**Figure 1 viruses-14-01890-f001:**
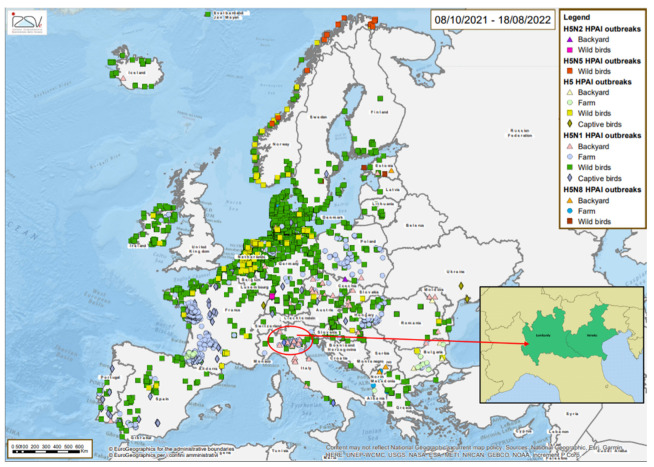
Map of the outbreaks detected during the HPAI 2021–2022 epidemic in Europe (source: https://www.izsvenezie.it/documenti/temi/influenza-aviaria/situazione-epidemiologica-europa-HPAI/2021-1/mappe.pdf, accessed on 24 August 2022). In the red circle the affected regions of northern Italy (Veneto and Lombardy).

**Figure 2 viruses-14-01890-f002:**
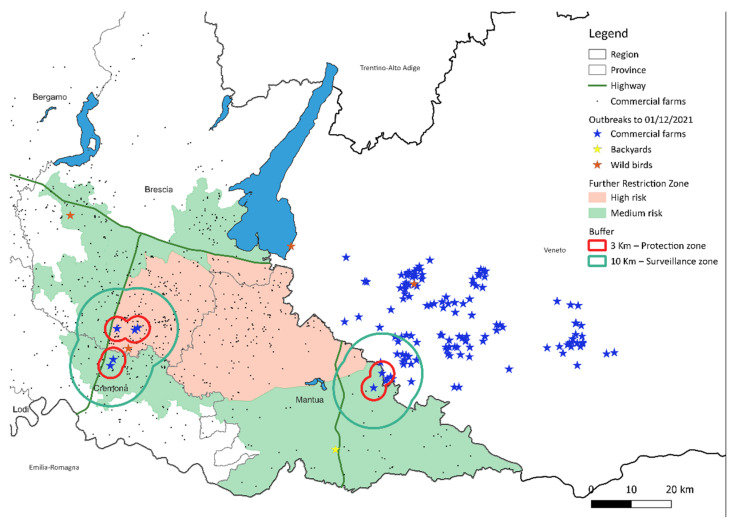
Map of outbreaks as at 1 December 2021 in Lombardy and in Veneto.

**Figure 3 viruses-14-01890-f003:**
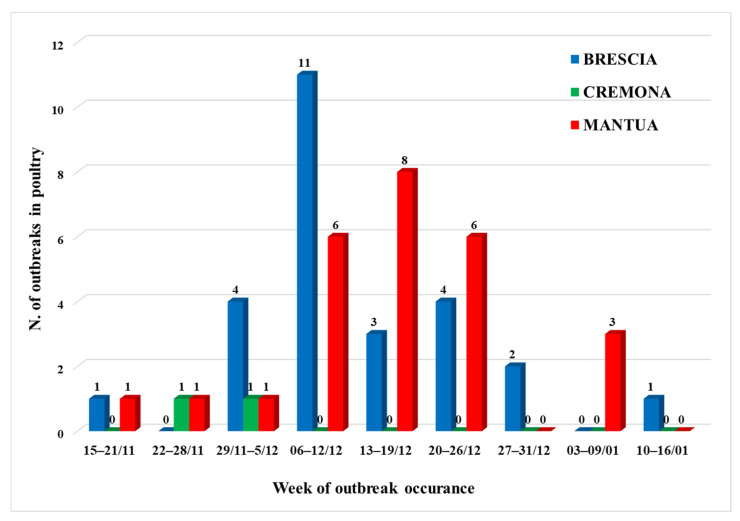
Distribution of outbreaks in industrial farms detected during the HPAI 2021–2022 epidemic in Lombardy, by week and by province.

**Figure 4 viruses-14-01890-f004:**
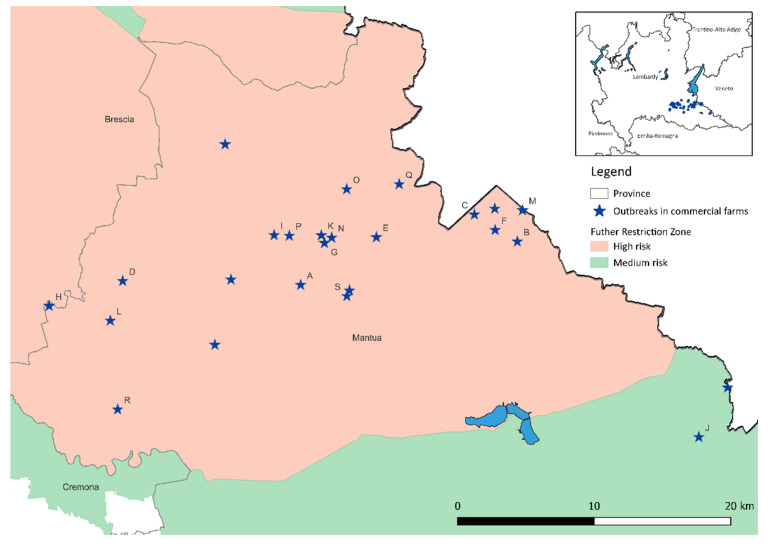
Map of high-risk contact between the HPAI outbreak sites in the province of Mantua.

**Figure 5 viruses-14-01890-f005:**
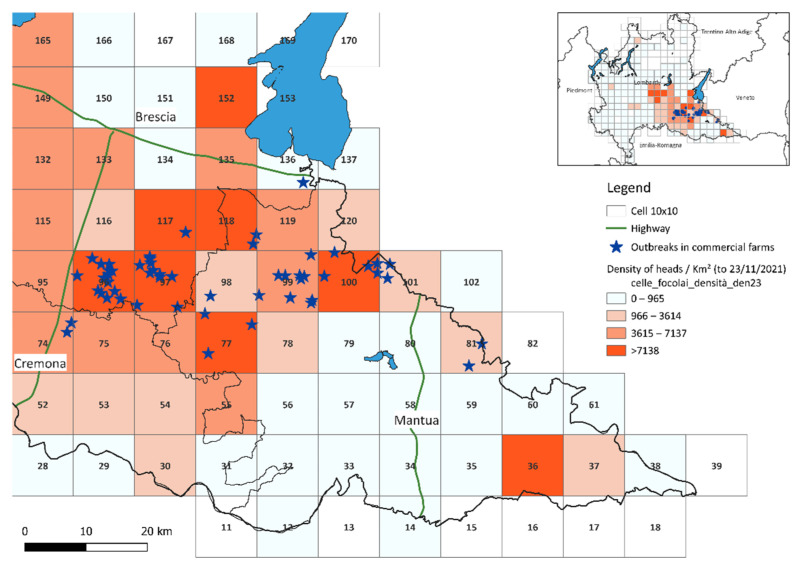
Map of avian influenza outbreaks and 10 × 10 km cells highlighted according to the density of birds per km^2^.

**Table 1 viruses-14-01890-t001:** Outbreaks detected during the HPAI 2021–2022 epidemic in the Lombardy region.

Province	Turkeys	Layers	Broilers	Ducks	Breeding Chickens	Guinea Fowl	Backyard	Wild	TOTAL
BRESCIA	5	11	6	3	1	0	1	2	29
MANTUA	14	9	2	0	0	1	1	1	28
CREMONA	1	0	1	0	0	0	1	0	3
PAVIA	0	0	0	1	0	0	1	0	2
COMO	0	0	0	0	0	0	1	0	1
BERGAMO	0	0	0	0	0	0	0	1	1
**LOMBARDY**	**20**	**20**	**9**	**4**	**1**	**1**	**5**	**4**	**64**

**Table 2 viruses-14-01890-t002:** Proportion of outbreaks in intensive farming affected by risk factors.

Risk Factors	% Outbreaks in Intensive Farming
Protection and surveillance zone	96%
Belonging to poultry companies	98%
Potential contact with wild birds	91%
Proximity of water sources	100%
Forced ventilation system	61%
Natural ventilation system	39%
Indirect contact	48%
via staff	7%
via vehicles in the province of Mantua	77%
reported by other region (Veneto)	4%

**Table 3 viruses-14-01890-t003:** High-risk contact between HPAI outbreak sites in the province of Mantua.

Vehicle	Outbreak 1	Number of Deliveries	Delivery Date	Date of Susp. Infection	Outbreak 2	Number of Deliveries	Delivery Date	Date of Susp. Infection	Outbreak 3	Number of Deliveries	Delivery Date	Date of Susp. Infection
1	A	2	7 December 2021	14 December 2021	M	2	15 December 2021	20 December 2021	**R**	1	29 December 2021	3 January 2022
2	B	1	30 November 2021	11 December 2021	G	2	07 December 2021	15 December 2021	**L**	1	15 December 2021	15 December 2021
3	**C**	1	7 December 2021	21 December 2021	A	1	11 December 2021	14 December 2021	**C**	1	18 December 2021	21 December 2021
4	D	3	7 December 2021	10 December 2021	N	1	14 December 2021	20 December 2021				
5	**E**	1	29 November 2021	5 December 2021	**F**	2	1 December 2021	10 December 2021				
6	**F**	2	1 December 2021	10 December 2021	**E**	1	1 December 2021	05 December 2021	**F**	3	2 December 2021	10 December 2021
7	G	1	2 December 2021	15 December 2021	O	1	3 December 2021	17 December 2021	**L**	1	10 December 2021	15 December 2021
8	**E**	1	3 December 2021	05 December 2021	**F**	2	6 December 2021	10 December 2021				
9	**H**	1	2 December 2021	9 December 2021	O	1	6 December 2021	17 December 2021	S	1	9 December 2021	15 December 2021
10	J	5	19 November 2021	22 November 2021	**E**	1	22 November 2021	5 December 2021				
11	K	1	3 December 2021	14 December 2021	I	2	10 December 2021	21 December 2021				
12	**L**	1	8 December 2021	15 December 2021	P	1	10 December 2021	20 December 2021	**L**	1	13 December 2021	15 December 2021
13	B	1	9 December 2021	11 December 2021	**Q**	2	14 December 2021	23 December 2021				
14	J	1	20 November 2021	22 November 2021	**E**	1	24 November 2021	5 December 2021				
**Vehicle**		**Outbreak 4**	**Number of Deliveries**		**Delivery Date**	**Date of Susp. Infection**	**Outbreak 5**	**Number of Deliveries**		**Delivery Date**	**Date of Susp. Infection**	
3		**R**	1		30 December 2021	3 January 2022						
7		P	1		15 December 2021	20 December 2021						
9		O	1		14 December 2021	17 December 2021	**Q**	1		20 December 2021	23 December 2021	

Outbreak sites found in high-density areas are highlighted in bold.

**Table 4 viruses-14-01890-t004:** Number of flocks and outbreaks in cells where there was at least one positive farm and the relative bird densities per km^2^.

CELL	No of Farms as at 23 November 2021	No Commercial Outbreaks	% Positivity	Species	No Birds/km^2^			
					23 November 2021	9 December 2021	30 December 2021	11 January 2022
96	63	14	22%	8 Layers 3 Broilers 2 Ducks 1 Turkeys	14,859.07	12,069.6	9110.8	8721.38
97	43	11	26%	5 Turkeys 2 Broilers 2 Layers 1 Breeding chickens 1 Ducks	9141.88	8254.82	5717.62	4073.1
99	34	10	29%	8 Turkeys 1 Layers 1 Broilers	6704.91	9060.8	5440.53	3995.7
100	17	5	29%	3 Layers 1 Guinea fowl 1 Turkeys	9281.31	5538.7	4412.1	3279.21
77	14	3	21%	3 Layers	8977.75	7285.63	7276.63	7276.63
74	22	2	9%	1 Broilers 1 Turkeys	5897.02	5434.2	2723.46	1722.3
81	7	2	29%	1 Layers 1 Turkeys	2430.16	2410.2	2296.2	2296.2
101	6	2	33%	1 Layers 1 Turkeys	2135.3	1370.67	1135.9	1135.9
118	17	2	12%	2 Turkeys	7764.74	7954.3	8258.58	7474.16
98	13	1	8%	1 Broilers	3613.88	2800.92	1630.62	1503.07
117	32	1	3%	1 Layers	9666.65	11,370.26	7874.12	5785.92
136	3	1	33%	1 Broilers	331.56	547.45	547.45	387.29

**Table 5 viruses-14-01890-t005:** Results of the statistical tests of the density classes of the birds per km^2^ as of 23 November 2021, 9 December 2021, 30 December 2021 and 11 January 2022.

	Cells with Outbreaks	Cells without Outbreaks	*p*-Value *
Density as at 23 November 2021	6347.5	950.6	0.0003
Density as at 9 December 2021	5608.5	943.7	0001
Density as at 30 December 2021	4370.5	899.4	0001
Density as at 11 January 2022	3694.5	903.6	0002

* Welch *t*-test for densities.

**Table 6 viruses-14-01890-t006:** Risk and probability that an outbreak will develop in the cells considered.

	Risk	95% CI	*p*-Value
**Density 23 November 2021**	1.00	1.0002–1.0006	<0.0001
**Probability of an outbreak occurring in a cell with a bird density per km^2^ of:**			
100	2.0%	0.7–5.4%	-
500	2.4%	0.9–6.1%	
1000	3.0%	1.2–7.1%	
5000	16.5%	9.2–27.6%	
10,000	66.7%	38.3–86.6%	

## Data Availability

Not applicable.

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
