# Peer review of "Epidemiological Features of the Highly Pathogenic Avian Influenza Virus H5N1 in a Densely Populated Area of Lombardy (Italy) during the Epidemic Season 2021–2022"

_viruses, 2022, doi:10.3390/v14091890_

Round 1

Reviewer 1 Report

1. The introduction only describes the epidemic status of H5N1 in Italy in detail, please describe the interaction with each other of H5N1 outbreak between Italy and other European countries, because it is difficult to draw a reliable pattern of H5N1 epidemics in a single country in Europe.

2. The impact of migratory birds on the occurrence of H5N1 in Italy even in Europe is not covered in this paper.

3. The is no direct experimental evidence described in this paper that the H5N1 strains circulating in Italy are highly pathogenic avian influenza virus.

4. What is the homology of H5N1 strains detected in different regions in Italy? Are they the same strain? Was the H5N1 strain in Italy being transmitted by migratory birds? or did it spread from other parts of Europe to Italy by other means?

5. The authors had given detailed description and summary of the H5N1 epidemic in Italy, why were the molecular characteristics of the H5N1 strains not examined?  Especially, the genetype of the detected H5N1 strains? Suggest add research work in this area.

6. Reference 4,6 and 7 have a far-fetched relationship with the content of the paper, replace or delete them please.

Author Response

We would like to thank the reviewers who with their comments helped to improve the quality of the manuscript. In attached there is the list of comments received with corrections made.

Reviewer 2 Report

The manuscript “Epidemiological features of the highly pathogenic avian influenza virus H5N1 in a densely populated area of Lombardy during the epidemic season 2021-2022” by Bellini and co-authors demonstrated and discussed risk factors associated with avian influenza outbreaks in Lombardy, Italy 2021-2022. The authors analysed available data, with no additional virological or genetic analysis. However, the following points should be considered and addressed by the authors in the manuscript prior to being submitted for publication as detailed below:

#Major:

-Introduction line 49-60: Is this part previously published or generated in this manuscript? Why not placed in the results section. It is the opinion of this reviewer that the structure of the manuscript should be revised. Same lines 78-81.

-Figure 1 and 2. should be revised: the authors showed outbreaks in two regions: Veneto and Lombardy. Colors are misleading and the authors showed different sectors in Lombardy only not Veneto.

#Minor:

-Title: I suggest that authors include “Italy” in the title as not many readers are familiar with the location of Lombardy.

-Line 18: “In cells….” What do the authors mean by cells?

-Line 19: “was higher than 66.7%” please mention the percentage in the low density area.

-Line 33: “Brescia 457, Mantua 207 and Cremona 173”. Clarify what are the numbers indicate?

-Line 37: add reference?

-Line 41: “H5N” is it highly or low pathogenic please clarify.

-Line 42: change “the disease” to “the virus”

-Line 43: change “highlighted” to “found or detected or reported”. Same line 45.

-Figure 1, 2: it is not easy to distinguish between “green and blue stars” and “orange and red stars” for many reads. I think this should be updated.

-Figure 1: is “backyard farm” something specially in Italy or you mean “backyards”

-Line 82: reference?

-Line 13: do you mean investigated/analyzed?

-Line 130: “we used 3 sources….” numbers below 10 should be written as words.

-Line 130 and 163: it is better two avoid “we, our…” in the whole text.

-Line 140: “HPAI” to “HPAI virus”. Please check in the whole text.

-Table 1: please indicates where are the farms? Broiler chicken?

-Line 170-171: please rephrase

Author Response

(The authors gave the same response as above.)

Round 2

Reviewer 1 Report

I agree with the author's reply.

Author Response

Thank you.

Reviewer 2 Report

The authors have addressed my concerns. However only two minor points should be fixed prior to acceptance. 
-Figure 1: please show where are Lombardy and in Veneto.

-Table 1: are turkey, duck, broiler, layers being part of farms (if so please indicate) or from backyard

Author Response

Dear Reviewer,

Figure 1 has been adjusted in accordance with the request.

In table 1, while for the backyards (5/64) the name of the column (before wild), has been kept generic because they are small family-run farms that have few animals but mixed species, e.g .: 3 chickens, a couple of ducks.. 1 Guinea fowl